# Evolution and Driving Factors of the Spatiotemporal Pattern of Tourism Efficiency at the Provincial Level in China Based on SBM–DEA Model

**DOI:** 10.3390/ijerph191610118

**Published:** 2022-08-16

**Authors:** Junli Gao, Chaofeng Shao, Sihan Chen

**Affiliations:** 1College of Environmental Science and Engineering, Nankai University, 38 Tongyan Road, Jinnan District, Tianjin 300350, China; 2Shenzhen Academy of Environmental Sciences, No. 50, Honggui 1st Street, Luohu District, Shenzhen 518000, China

**Keywords:** tourism efficiency, spatiotemporal patterns, panel estimates, driving factors

## Abstract

In order to give guidance to improve tourism competitiveness and sustainable development, it is particularly important to identify and analyze the factors and mechanisms that affect efficiency. The SBM–DEA model including undesirable outputs was used to measure the tourism efficiency of 30 provinces in China from 2006 to 2019. Combined with the compound DEA model, the sensitivity of each province to the fluctuation of the input–output index was mined. The exploratory spatial analysis method and fixed effect model were used to analyze the spatial change and driving factors of tourism efficiency. The results show that: (1) the tourism efficiency of each province in China fluctuated from 2006 to 2019, and the average value was raised from 0.12 to 0.71, generally reaching the grade of medium and high efficiency; (2) the spatial difference of tourism efficiency is significant, but there is no obvious spatial correlation; (3) the most important input factors to tourism efficiency are environmental resources, tourism resource inputs and tourism infrastructure construction, and tourism fixed asset investment is redundant. (4) Optimizing the industrial structure, strengthening the introduction of core technology, and continuously promoting the process of urbanization and marketization are important ways to improve the efficiency of tourism.

## 1. Introduction

In the past few decades, tourism has accounted for 10% of global GDP and jobs, and is considered to be an important driver of economic growth with the increase in per capita disposable income [1]. Tourism was once called a “smokeless industry”. However, the links of transportation, catering, accommodation, sightseeing and entertainment in tourism activities not only consume a lot of energy and water resources, but also produce a lot of air pollutants such as sewage, solid waste and carbon dioxide [2], thus affecting the sustainable use of landscape resources, affecting the transformation of China’s tourism from the stage of rapid growth to the stage of high-quality development [3].

For any economic entity, improving efficiency is an important condition to maintain its sustainable development [4]. Tourism efficiency has gradually become one of the hot spots in tourism research in recent years [5]. The current research is mainly divided into tourism industry factor efficiency, tourism economic efficiency and tourism ecological efficiency. Different research fields of tourism efficiency involve different tourism destination scales such as countries, provinces, counties and cities, and villages, and mainly focus on the differences in spatial and temporal patterns, evolution characteristics, driving mechanisms and factors affecting efficiency. The non-parametric method discusses the efficiency indicators of pure technical efficiency and scale efficiency, and further analyzes the coupling level of tourism efficiency, tourism development scale and economic development level from a macro perspective, and analyzes the internal state of the tourism system from a micro perspective [6]. The research on the factor efficiency of the tourism industry mainly focuses on the factors of the tourism industry, such as travel agency, tourism scenic spot, tourism accommodation industry, tourism transportation industry and so on [7]. Most of them are carried out from the enterprise level. Enterprises are an important force that cannot be ignored in the development of tourism industry, environmental protection and technological innovation [8]. The economic efficiency of tourism is a comprehensive performance measurement for the development of one or more tourism destination units. When considering the maximization of input and output, economic indicators are selected, such as fixed asset investment, actual use of foreign investment and tourism income [9,10,11]. Tourism eco-efficiency is obtaining the maximum economic output with the least resource consumption and environmental cost in tourism activities, taking into account the input of ecological environment resources and the undesired output of environmental pollution caused by tourism [12,13,14]. With the passage of time, the study of tourism efficiency has gradually developed to more complex topics, such as influencing factors, spatial effects, driving factors and the sustainable development of tourism. Guo et al. (2022) [15] combined tourism efficiency with new geographical technology and used a geographic detector model to determine the determinants related to the spatial differentiation of tourism ecological efficiency. Some studies have explored the spatiotemporal characteristics of tourism efficiency through exploratory spatial data analysis. Li et al. (2020) [16] adopted a spatial autocorrelation model, which showed that the overall spatial correlation of tourism efficiency in the Wuling Mountain area of China was weak. Some scholars have carried out research on tourism ecological security, which is an important direction for the sustainable development of tourism destinations. Liu et al. (2022) [17] assessed the spatiotemporal evolution and dynamic characteristics of the ecological security of tourism, as well as the influencing factors and driving mechanisms. In addition, Ma et al. (2022) [18] proposed a new framework for measuring the temporal and spatial distribution of cross-provincial tourism demand based on search engine index from a geographical perspective, which combines spatial autocorrelation and Geo detector to identify the spatiotemporal distribution pattern of tourism demand at the provincial level in China and detect its driving mechanisms.

Although the study of tourism ecological efficiency has considered the undesirable output of environmental pollution categories such as tourism carbon emissions, most of them consider the undesirable output narrowly, ignoring social input and social output [19,20,21]. It is not in line with the vision of economic, social, environmentally sustainable development and high-quality development of tourism destinations. Although previous studies on the determinants of tourism efficiency have made substantial contributions to tourism literature, little attention has been focused on the external macrodeterminants associated with efficiency, such as economic, social, technical, and institutional factors. At the same time, most of the perspectives of these studies focus on the analysis of specific cases, and they are relatively simple in terms of research scale. In summary, domestic and foreign research results provide certain theoretical support and reference for further research on tourism efficiency, but there is still room for improvement in related research. This paper focuses on the current demand for the high-quality development of tourism, introduces the input and output of economy, society and environment into the study of tourism efficiency, on the basis of comprehensively considering the undesirable outputs such as tourism environmental resource inputs and tourism carbon emissions, tourism water pollution emissions and tourism solid waste production; the social benefits brought by tourism are also taken into account.

Based on this, the input–output index system is constructed, and the SBM–DEA model based on undesirable outputs, the compound DEA model and fixed effects model are used to measure China’s provincial tourism efficiency, identify the key indicators of input and output, and analyze the spatial differentiation law of tourism efficiency by exploratory spatial data methods, further exploration of the drivers of tourism efficiency. This paper analyzes the driving mechanism of tourism efficiency from 10 aspects such as regional economic level, quality of fiscal revenue, level of opening to the outside world, degree of marketization, urban and rural structure, level of digitization, location traffic status, incidence of public health events and technological innovation investment, and selects key indicators to measure the main driving factors of tourism efficiency.

## 2. Materials and Methods

### 2.1. Data and Study Area

In this study, 22 provinces, 4 municipalities directly under central government (Beijing, Tianjin, Shanghai, and Chongqing), and 4 autonomous regions (Inner Mongolia, Ningxia, Xingjiang, and Guangxi) in China were selected as study areas; we use the word “province” to refer to these study areas throughout the rest of the paper. Furthermore, because the date for Tibet, Taiwan, Hong Kong, and Macao were unavailable, these provinces were excluded. In this paper, we collected the indicator data of China’s provinces for 14 consecutive years from 2006 to 2019, measured their long-term tourism efficiency, and used a panel regression model to reveal the driving factors of tourism efficiency.

The data mainly came from the “China Statistical Yearbook”, “China Cultural Relics Statistical Yearbook”, “China Tourism Statistical Yearbook (original and duplicate)”, “China Urban Construction Statistical Yearbook”, “China Population and Employment Statistical Yearbook”, “China Environmental Statistical Yearbook”, Provincial Statistical Yearbooks, lists, directories and other statistical data published on the official website. Missing data were filled by national averages, adjacent year averages, linear interpolation, trend extrapolation and other methods.

### 2.2. Methods

#### 2.2.1. SBM–DEA Model Based on Undesirable Output

DEA (data envelopment analysis) is a linear programming method for evaluating relative effectiveness proposed by Charnes in 1978 [22]. It is a non-parametric evaluation method for complex decision-making units with multiple input and output indicators. It can be applied to the measurement of tourism efficiency of input–output index system involving multi-factor integration. At the same time, DEA is also the focus of international tourism efficiency research [23]. The SBM (Slacks-Based Measure)-undesirable model was put forward by Tone on the basis of the traditional DEA model [24] to improve the defect of the radial aspect and introduce the unexpected output index to make up for the measurement deviation of the radial direction and angle of the DEA model. It was used to measure tourism efficiency including the unexpected output of tourism pollution emissions in this paper. The details are as follows: suppose there are *N* decision-making units (DMU*j*, *j* = 1, 2, 3, …, *N*), each DMU has *n* kinds of inputs *X*, *m*_1_ kind of expected output *Y^a^* and *m*_2_ kind of expected output *Y^b^*. The vector expression is *x*∈*R^n^*, *y^a^*∈*R^m^*_1_, *y^b^*∈*R^m^*_2_. The input-output matrix can be defined as: $X=\left[x_{1}{,\;x}_{2}{,\;\ldots ,\;x}_{N}\right]\;\in {\;R}^{N\times n}$,$\;Y^{a}=\left[y_{1}^{a},y_{1}^{a},\ldots ,y_{N}^{a}\right]\in R^{N\times m_{1}}$,$\;Y^{b}=\left[y_{1}^{b},y_{1}^{b},\ldots ,y_{N}^{b}\right]\in R^{N\times m_{2}}$, $x_{i}$ > 0, $y_{i}^{a}$ > 0, $y_{i}^{b}$ > 0. For the linear programming expression of the SBM model with variable returns of scale, see expressions (1) and (2):(1)$$\min\rho =\frac{1\;-\;\frac{1}{n}\sum_{i=1}^{n}\frac{S_{i}^{-}}{x_{i0}}}{1+\frac{1}{m_{1}{+m}_{2}}\left(\sum_{r=1}^{m_{1}}\frac{S_{r}^{a}}{y_{r0}^{a}}+\sum_{r=1}^{m_{2}}\frac{S_{r}^{b}}{y_{r0}^{b}}\right)}$$
(2)$$s.t.\left\{\begin{array}{c} x_{i0}{=X\lambda +S}_{i}^{-}\\ {\;y}_{r0}^{a}{=Y}^{a}{\lambda \;-\;S}_{r}^{a}\\ y_{r0}^{b}{=Y}^{b}{\lambda +S}_{r}^{b}\\ S_{i}^{-}\;\geq {\;0,\;S}_{r}^{a}\;\geq {\;0,\;S}_{r}^{b}\;\geq \;0,\;\lambda \;\geq \;0 \end{array}\right.$$

In the formula, *S* represents the input slack variable; *S^a^* represents the expected output slack variable; *S^b^* represents the undesired output slack variable; λ represents the weight vector. When *ρ* = 1, *S*^−^ = 0, *S^a^* = 0, *S^b^* = 0, the comprehensive technical efficiency, pure technical efficiency and scale efficiency of the decision-making unit are relatively effective. When 0 ≤ *ρ* < 1, it means that the decision-making unit is relatively inefficient. This paper draws on other studies and uses the equidistant division method to divide the obtained efficiency values into four grades, namely: low efficiency (0 ≤ *ρ* < 0.25), medium efficiency (0.25 ≤ *ρ* < 0.50), medium and high efficiency (0.50 ≤ *ρ* < 0.75) and high efficiency (0.75 ≤ *ρ* < 1) grade.

#### 2.2.2. Composite DEA Model

The composite DEA method can repeatedly adjust the input and output indicator system, compare different results, and identify indicators that have a significant impact on the effectiveness of the DMU [25,26]. The main points of the composite DEA method can be summarized as: for a given set of decision-making units and a set of evaluation indicators *D*, select *D_i_* (*i* = 1, 2, …, *τ*), such that *D* ⊃ *D_i_*, and use the appropriate DEA model to find the effectiveness coefficient vectors *ρ*(*D*), *ρ*(*D*_1_), …, *ρ*(*D_τ_*) related to each index set, and use these vectors as variables to establish the function *F = F*(*ρ(D*_1_), …, *ρ*(*Dτ*)). There are two main modes used in this article:

(1) Analyze the influence of a certain index on the decision-making unit in the evaluation system. *D_i_* represents the index system after removing the *i*th evaluation index in *D*, and *ρ*(*D*) and *ρ*(*D_i_*) can be obtained. The definition is as follows:
(3)$$S_{j(i)}=\frac{\rho_{j}\left(D\right){\;-\;\rho }_{j}\left(D_{i}\right)}{\rho_{j}{(D}_{i})},\;j=1,\;2,\;\ldots ,\;n$$

If the decision unit *j*_0_ satisfies: $S_{j0}\left(i\right)=max\left\{S_{j}\left(i\right)\right\},\;j=1,\;2,\;\ldots ,n$, *j* = 1, 2, …, *n*, it shows that the decision-making unit *j*_0_ has a relative advantage in terms of input or output compared to other decision-making units in terms of *i*-th index, because its relative efficiency value increases the most after adding *i*-index. If it is an input indicator, it may also indicate that the input of this indicator is seriously insufficient, which has a significant impact on the relative efficiency of input and output.

(2) Analyze the reasons for the formation of an invalid unit. If a decision-making unit *j*_0_ is non-DEA valid under the *D* index, the definition is as follows:
(4)$$S_{i}=\frac{\rho_{j0}\left(D\right){\;-\;\rho }_{j0}\left(D_{i}\right)}{\rho_{j0}{(D}_{i})},\;i=1,\;2,\;\ldots ,\;t$$

Taking *i*_0_ to make $S_{i0}{=\min(S}_{1},\;S_{2}{,\;\ldots ,\;S}_{t})$ indicates that index *i*_0_ is the index that has the greatest influence on the ineffectiveness of decision-making unit *j*_0_, probably because the input corresponding to this index is redundant and the utilization rate is too low, or the output efficiency corresponding to this index is too low to reach the effective output scale.

#### 2.2.3. Exploratory Spatial Analysis

Using the ESDA (exploratory spatial data analysis) method [27], global Moran’s I index and local Moran’s I index were used to measure the spatial correlation to measure the spatial distribution pattern of tourism efficiency at the provincial level in China. The global spatial autocorrelation coefficient Moran’s I was used to measure the spatial distribution characteristics of the tourism efficiency of each province in the entire study area, and the local spatial autocorrelation coefficient Local Moran’s I was used to explore the degree of correlation between the tourism efficiency of the study area and the same attribute of its neighboring locations in sub-regions [28,29]. The methods of global autocorrelation Moran’s I and local autocorrelation Local Moran’s I are as follows:(5)$$\mathrm{Moran}’s\;I=\frac{n\sum_{i=1}^{n}\sum_{j=1}^{n}w_{\mathrm{ij}}{(x}_{i}-{\bar{x})(x}_{j}-\;\bar{x})}{(\sum_{i=1}^{n}\sum_{j=1}^{n}w_{\mathrm{ij}})\sum_{i=1}^{n}(x_{i}-\;\bar{x}\;{)}^{2}}$$
(6)$${\mathrm{Local}\;\mathrm{Moran}’s\;I}_{i}=\frac{x_{i}-\bar{x}\;}{\sum_{i}{{(x}_{i}-\;\bar{x}\;)}^{2}}\underset{j}{\sum }w_{\mathrm{ij}}{(x}_{j}-\;\bar{x})$$

In the formula, *n* is the total number of study areas, *x_i_* and *x_j_* are the tourism efficiency of study areas *i* and *j* respectively, $\bar{x}$ is the average efficiency of all study areas, *w_ij_* is the spatial weight matrix between *i* and *j*, and *s* is the standard deviation. In this paper, the distance function was used to calculate the spatial weight of each province.

#### 2.2.4. Panel Regression Model

Generally speaking, three models can be established for the analysis of short panel data: mixed regression model, fixed effect model and random effect model [30]. The basic assumption of the mixed regression model is that there is neither individual effect nor time effect in the panel data:(7)$$y_{\mathrm{it}}{=x}_{\mathrm{it}}^{\prime }{\beta +u}_{\mathrm{it}},\;i=1,\;\ldots .,\;N,\;t=1,\;\ldots ,\;T$$

When the individual effect is added to the mixed regression model, the model is transformed into a fixed effect model:(8)$$y_{\mathrm{it}}{=\gamma }_{i}{+x}_{\mathrm{it}}^{\prime }{\beta +u}_{\mathrm{it}},\;i=1,\;\ldots ,\;N,\;t=1,\;\ldots ,\;T$$

In the formula, *i* stands for individual, $\gamma_{i}$ is individual effect, and $u_{it}$ is called the characteristic error term or trait disturbance term. In the fixed utility, $\gamma_{i}$ is regarded as an unobserved random variable related to $x_{it}$, which can be estimated one by one, which is called fixed effect.

Assuming that $\gamma_{i}$ is a random variable independent of $x_{it}$, the model is called a random effect model. The following auxiliary assumptions are usually required:(9)$$\gamma_{i}\sim (\gamma ,\;\sigma_{r2}){,u}_{\mathrm{it}}\sim (0,\;\sigma_{u2})$$

In the selection of panel models, the F test is generally used to select the mixed regression model or the fixed effect model, the LM test is used to select the mixed regression model and the random effect model, and the Hausman test is generally used for the selection of the fixed effect model and the random effect model.

### 2.3. Construction of Tourism Efficiency Measurement Index

The tourism efficiency referred to in this paper aims to obtain higher tourism economic, social output or service value with as little tourism economic and social input and as low environmental loss as possible, and to reduce the emission of environmental pollutants in the process of tourism as much as possible. On the basis of previous studies and fully considering the needs of sustainable development of tourism, the input–output indicators constructed were integrated with the three pillars of society, economy and environment emphasized in the United Nations Sustainable Development goals (SDGs) [31]. The relevant target indicators of SDG6, 7, 8, 10, 11, 12 and 15 which are directly or indirectly related to tourism development are mainly considered. Based on this, in this paper, we constructed a tourism efficiency measurement index including 5 input indicators and 4 output indicators, which meets the requirement that the number of elements in the DEA reference set is not less than three times the total number of input and output indicators. Table 1 shows the indicators of tourism efficiency at the provincial level in China.

## 3. Results

### 3.1. Analysis of the Overall Characteristics of Tourism Efficiency

The change of TE value, TE mean value and grade quantity distribution of each province was calculated according to the SBM-undesirable-DEA model, as shown in Figure 1. As can be seen from Figure 1, from 2006 to 2019, the average value of China’s TE increased from 0.12 to 0.71, and the overall efficiency level changed from low efficiency level to medium-high efficiency level. The development trend of the national TE average includes two stages: the first stage is from 2006 to 2016, in which the average efficiency increased slowly, and the overall efficiency level changed from low efficiency level to medium efficiency level; the second stage is from 2016 to 2019, in which the average efficiency increased rapidly, and the overall efficiency level changed to the middle and high efficiency level, and was close to the high efficiency level. From 2006 to 2019, the TE level of all provinces increased, the proportion of provinces at high efficiency level increased from 0 to 47%, the proportion of provinces at medium and high efficiency level increased from 0 to 10%, the proportion of provinces in the medium efficiency level increased from 10 to 37%, the proportion of provinces at low efficiency level decreased from 90 to 7%, and the number of provinces with full efficiency increased from 0 to 47%. In 2019, Tianjin, Hebei, Shanxi, Inner Mongolia, Jilin, Henan, Guangxi, Chongqing, Guizhou and other provinces reached full effectiveness.

The change trend of TE mean in eight regions of China from 2006 to 2019 is shown in Figure 2. As can be seen from Figure 2, the average development trend of TE in the eight regions was consistent with that of the whole country. The average value of TE in China increased by 0.58. The average increase in TE in the eight regions ranged from 0.25 to 0.72. Among them, the growth rate of TE in the middle Yellow River, the southwestern region, the northeast region and the eastern coastal region was higher than the average increase in China, and the increase in the middle Yangtze River and the southern coastal areas was the lowest. According to relevant research, the high-quality development level of China’s economy showed a spatial distribution pattern of “high in the east and low in the west” and “high in the south and low in the north”, and gradually decreased “from south to north” and “from east to west” [37]. From the above conclusions, it can be seen that the spatial distribution pattern of China’s TE is different from that of China’s economic pattern, indicating that economic factors are not the main driving factors of TE. Therefore, it was necessary to further explore its driving factors.

### 3.2. Analysis of the Results of Compound DEA of Tourism Efficiency

Through the lateral analysis of the composite DEA results of the non-effective DMU, the reasons and optimization directions of the non-DEA effectiveness of TE in each year can be obtained. Specifically, the coefficient *S* of all input indicators in Sichuan and Hunan was greater than 0, indicating that there was no redundancy in the input indicators of these two provinces, but there was a problem of low utilization of tourism fixed assets and tourism resources. The coefficient *S* of tourism fixed assets and tourism employees was close to 0, indicating that the investment efficiency of tourism fixed assets and employees in these two provinces was too low; Jiangxi and Hunan had too much investment in tourism employees and tourism resources; too much investment in tourism fixed assets and tourism resources in Guangdong, Hainan and Yunnan; Liaoning, Zhejiang, Shandong, Hubei and Xinjiang also had problems of excessive investment in tourism fixed assets and insufficient utilization; Beijing had the problem of redundant investment in tourism fixed assets and excess personnel in the tourism industry; the main factor of Heilongjiang’s non-DEA effectiveness was the insufficient utilization of tourism resources; the main problem was the redundant number of employees in Shaanxi; except for tourism resources and environmental resources in Gansu, the rest of the input was redundant.

By analyzing the cumulative value of the evaluation results of compound DEA, we could identify the indicators that have a significant impact on the effectiveness of tourism efficiency in various provinces. In terms of input indicators, at the national level, the cumulative value of *S_j_*(*i*) was mostly negative before 2010 and positive after 2010, indicating that the phenomenon of resource redundancy and low resource utilization has been significantly improved over time. Specifically, the cumulative value of $\sum S_{j}\left(5\right)$ was much higher than that of the other four inputs, followed by $\sum S_{j}\left(4\right)$, $\sum S_{j}\left(2\right)$ and $\sum S_{j}\left(1\right)$, $\sum S_{j}\left(3\right)$ had the lowest cumulative value, which shows that the most important input factors to TE at the national level are environmental resources input, tourism resources input and tourism infrastructure construction. The most redundant investment index was the investment of tourism fixed assets. The cumulative value of *S_j_*(*i*) in each region is shown in Figure 3 (I–Ⅷ represent the Northern coast, Northeast, Eastern coast, Southern coast, Middle Yellow River, Middle Yangtze River, Southwest and Northwest, respectively). As can be seen from Figure 3, the input indicators with higher cumulative value in most regions were consistent with the national level, and the investment indicators with the lowest cumulative value in most regions were tourism fixed assets. Except for this index, most of the other investment indicators with lower cumulative value were tourism employees. The cumulative value of *S_j_*(*i*) of the output index was significantly higher than that of the input index. At the national level, $\sum S_{j}\left(6\right)$ and $\sum S_{j}\left(7\right)$ were the highest, indicating that the relative efficiency of national tourism input and output has an impact on total tourism revenue and the total number of tourists was the most sensitive, and $\sum S_{j}\left(8\right)$ was the lowest, indicating that the urban–rural income gap index had the smallest contribution to the relative efficiency of tourism. The output indicators with the highest cumulative value in most regions were also total tourism receipts and total tourist arrivals.

### 3.3. Spatial Correlation Analysis of Tourism Efficiency

Based on the geographic weight matrix, according to the method of spatial correlation test, Stata16.0 software (StataCorp LLC, College Station, TX, USA) was used to measure the Moran’s I index of TE. The results are shown in Table 2. It can be seen from Table 2 that the Moran’s I index of TE was negative in most years in the study period, and the absolute value was very small, and the P value was very insignificant, failing to pass the 1% significance level test. It shows that there was no obvious spatial correlation between TE in each province during the study period, and the TE in each region was basically independent and not affected by neighboring regions.

The Moran scatter plot of TE from 2006 to 2019 was drawn to further study the spatial agglomeration characteristics of tourism efficiency, and then judge the local spatial correlation of TE in each province. Due to limited space, the main excerpts of the four years are shown in Figure 4. It can be seen from Figure 4 that there were two main spatial aggregation states of TE in 2006, 2010 and 2015, namely the high-low aggregation state in the second quadrant and the low-low aggregation state in the third quadrant. With the change of time, the provinces fell more evenly in the four quadrants of the Moran scatter chart, and the proportion of provinces falling in the first to fourth quadrants in the 2019 Moran scatter chart was 20, 30, 20 and 20%, respectively, indicating that their spatial correlation was further reduced.

### 3.4. Panel Regression Result Analysis

Tourism development has multi-dimensional characteristics, and the driving factors of tourism efficiency are also various. Drawing lessons from the existing studies [38,39,40,41,42,43,44] and considering the availability of data, in this paper, we examined its impact on TE from ten aspects It is worth noting that the main tourist attractions of tourism destinations are landscape resources including natural and cultural landscapes, and the endowments of landscape resources in different provinces are different, and the corresponding input, development and utilization of landscape resources and the orientation of tourism development are also different. At the same time, the endowment characteristics of landscape resources also determine which state different tourism destinations are in, such as “high input-high output”, “high input-low output” or “low input-high output”. However, as there is no more scientific and reasonable method to measure the endowment of different types of landscape resources, the endowment of landscape resources has not been included in the driving factors of this paper. The selection of factors affecting tourism efficiency and descriptive statistics are shown in Table 3.

In order to avoid data fluctuation, logarithmic processing of gdppc and tnd was carried out [46]. The 14-year lngdppc, pti, str, pieg, mi, ul, ip, lntnd, iid and rdi of each province were taken as explanatory variables and TE as explained variable to estimate the econometric model. The analysis results came from Stata16.0.

The results of the multicollinearity test show that the coefficient of variance expansion (VIF) was less than 10, that is, there is no multicollinearity in the explanatory variables, so the follow-up operation could be carried out directly. First of all, the F-test was used to compare the mixed regression model with the fixed-effect model, and the F-test strongly rejected the hypothesis that there is no difference between the mixed-effect model and the fixed-effect model, so the fixed-effect model should be accepted. Secondly, using LM test to compare mixed regression model and random effect model, LM test strongly rejected the original hypothesis that “there is no individual random effect”, so we should accept random effect model. Finally, the fixed effect model and the random effect model were compared, and the Hausman test suitable for heteroscedasticity and cross-sectional correlation was used to compare. The test results significantly rejected the random effect model, so in this paper, we selected the panel fixed effect model to analyze the driving mechanism of provincial tourism efficiency in China.

Before the regression analysis, based on the heteroscedasticity and autocorrelation attributes of the sample, we chose the nonparametric covariance matrix estimator proposed by Driscoll and Kraay. The estimation model can obtain the consistent standard error considering heteroscedasticity and autocorrelation, and is suitable for the fixed effect model. The results are shown in Table 4. In this paper, the above method was also followed to analyze the driving factors of TE in eight regions by panel regression analysis. Limited to space, the complete results are not listed, and only the significant driving factors in the analysis are discussed.

As can be seen from Table 4, the proportion of tertiary industry and urbanization rate coefficients in the explanatory variables was positive, and passed the significance test of 1%, indicating that when other factors remain the same, the improvement of the proportion of tertiary industry and the urbanization rate in various regions of China will significantly promote the improvement of tourism efficiency. The marketization index coefficient was positive and passed the significance test of 10%, indicating that while other factors remain unchanged, the marketization index of all provinces in China will increase by 1 percentage point, which will increase TE by an average of 0.0020 percentage points. The proportion coefficient of total import and export of goods to GDP was negative, which may be due to the vast territory of China, the impact of regional tourism development on foreign exchanges is mostly limited to the relatively developed areas of local economic level, and the transfer or transmission from abroad to our provinces is often not the core technology, which leads to the low ability of our provinces to digest and absorb foreign advanced technology, resulting in the ineffectiveness of China in the introduction of international developed technology.

The panel regression results of the provinces in the eight regions were as follows: the lntnd coefficient representing the traffic network density in the northern coastal area was positive (0.1394 *), indicating that the convenient traffic conditions in the northern coastal area were the main factor to improve its tourism efficiency; the rdi coefficients of the eastern coastal and northeast regions were both positive (0.3450 and 0.0759 *), indicating that the investment in technological innovation in these two regions could effectively improve their tourism efficiency; the urbanization rate coefficient of the middle Yangtze River and the northwest region was positive (0.0198 **, 0.0265 ***), indicating that the level of urbanization was the main factor affecting the tourism efficiency of these two areas; at the same time, the high proportion of tertiary industry structure was also the main factor for the improvement of tourism efficiency in the middle reaches of the Yangtze River (0.0098 **); the mi and pieg coefficients of the middle Yellow River were negative (−0.0701 *, −0.0162 *), indicating that the marketization and opening up level of the region had not played a positive role in promoting tourism efficiency, the lngdppc coefficient of the variable representing GDP per capita was positive (0.4216 ***), indicating that economic factors were also a strong driving factor for the improvement of tourism efficiency in the middle Yellow River; the pieg coefficient of Southwest China was negative (−0.0079 *) and the str coefficient was positive (0.097 *), indicating that the stable fiscal and taxation structure in the southwest region brought obvious positive spillover effects to the tourism development in this region; The coefficients of pti in the southern coastal areas was positive (0.0222 **), and different from the negative proportion coefficient of the total import and export of other provinces and other goods, the coefficient of this influencing factor in the southern coastal areas was positive (0.0016 **), indicating that the southern coastal areas have a strong ability to introduce, absorb and transform foreign advanced technology.

## 4. Discussion

### 4.1. Further Discussion

The World Economic Forum (WEF) has issued the Tourism Competitiveness report regularly since 2007 (consecutive annual reports in 2007, 2008 and 2009, and changed to a biennial edition since 2011, the Tourism Competitiveness Index consists of four pillars: supporting environment, tourism policies and conditions, infrastructure and natural and cultural resources) [47]. The score of China in these 8 years was standardized in the range of 0 to 1, and the correlation analysis with the provincial average TE in the corresponding year showed that there was a significant strong correlation between them (Spearman correlation coefficient is 0.905 times *). Big data Research Institute of Chinese Culture and Tourism issued the China Ecotourism Development Index report (composed of economic ecology, political ecology, cultural ecology, social ecology, natural ecology with 5 subsystems, 24 indicators) [48]. The correlation analysis between provincial ecotourism development index and provincial average TE showed that there was a weak correlation between them (Spearman correlation coefficient: 0.414 *).

Since 2020, various provinces in China have issued the Special Plan for Culture and Tourism Development in the 14th five-year Plan one after another. The documents of the vast majority of provinces have proposed to promote the integrated development of cultural tourism and speed up the quality, efficiency, transformation and upgrading of tourism. It can be seen that the integrated development of the culture and tourism industry is an important direction for China’s tourism development for a long time in the future. The report on the Integrated Development Index of Chinese Culture and Tourism focuses on the new hot spots of the integrated development of domestic culture and tourism. The integrated development index of culture and tourism is composed of four comprehensive layers and 43 specific indicators, which are composed of basic conditions, innovative ability, industrial level and social effects, and the current situation of the integrated development of culture and tourism in 31 provinces in China is evaluated and analyzed [49]. The correlation analysis between the provincial culture and tourism integration development index from 2015 to 2019 and the corresponding provincial TE mean value shows that there is no obvious correlation between them (Spearman correlation coefficient: 0.143), indicating that the development degree of culture and tourism integration in provinces with higher TE may not be higher.

From the point of view of linking up with the existing evaluation, the tourism efficiency results found in this paper have a significant positive correlation with tourism competitiveness index and eco-tourism development index, which shows that the results of this paper are scientific. From the point of view of connecting with the orientation of future development planning, provinces should consider the maximization of tourism efficiency on the basis of further promoting the integrated development of the culture and tourism industry. To sum up, this paper measures tourism efficiency by input–output indicators covering all aspects of tourism development, such as economy, society and environment. It avoids the narrow consideration of input–output indicators in previous studies and does not fully consider the input of tourism environmental resources and the unexpected output of environmental pollution. At the same time, it is scientific.

### 4.2. Policy Implications

Tourism efficiency varies from province to province in China. For provinces that have achieved high efficiency, such as Tianjin, Chongqing and Shanghai, measures should be taken to stabilize the achievements of tourism so as to further promote the integrated development of the culture and tourism industry. For the provinces that have not yet achieved high efficiency, we should pay attention to the causes of their ineffectiveness, and determine the optimal investment quota of tourism fixed assets in each region by means of prior evaluation, so as to reduce the redundant phenomenon of tourism fixed asset investment. Measures should be taken to improve the labor productivity of tourism employees and avoid the phenomenon of labor redundancy.

The driving factors of tourism efficiency are also different in different provinces, so the relevant countermeasures and measures to promote tourism development should also have their own emphasis. On the whole, optimizing the industrial structure, strengthening the introduction of core technology, and continuously promoting the process of urbanization and marketization are important ways to improve tourism efficiency. Tourism can promote not only the development of related industries, but also the development of surrounding areas. In view of the fact that there is no obvious spatial correlation in tourism efficiency among China’s provinces, China’s provinces should be based on different cultural and natural resource advantages. We should make full use of the advantages of convenient transportation and tourism brands to deepen inter-regional tourism cooperation and promote the coordinated and high-quality development of tourism and give full play to the radiation-driven role of provinces with high tourism efficiency to the surrounding areas.

### 4.3. Limitations and Future Research Direction

This paper takes the provincial level as the basic spatial research unit. Future research can expand the research perspective to the level of cities, counties or urban agglomerations, which can provide more valuable references for sustainable tourism development and global tourism. In addition, it is recommended that the national statistics department further refines the statistical caliber of tourism-related data, collects tourism-related statistical data, accelerates the establishment of a tourism satellite account, and supports comprehensive research on the sustainable development of China’s tourism.

In addition, the emergence of COVID-19 in December 2019 affected more than 200 countries and regions, and had an impact on industries such as health, biomedicine, environment, and tourism. Among them, hotels and tourism activities were production industries that were severely affected by the epidemic. Therefore, based on detailed data in the future, it is possible to quantitatively analyze the intensity of the impact of COVID-19 on tourism efficiency, explore its impact mechanism on tourism efficiency, and further explore how to improve crisis management capabilities and social governance levels in the face of emergencies.

## 5. Conclusions

In this paper, we evaluated the dynamic evolution characteristics, influencing factors and driving mechanism of tourism efficiency at provincial level in China for 14 consecutive years from 2006 to 2019. The main conclusions are as follows:(1)From the perspective of time, the TE values and grades of all provinces in China showed a fluctuating upward trend from 2006 to 2019. The average value of TE in all provinces increased from 0.12 to 0.71, and the provinces that were fully effective increased from 0 to 47%, reaching the level of medium-to-high efficiency as a whole.(2)From the spatial point of view, the spatial differentiation of TE in different provinces of China is significant, and there are two main spatial agglomeration states: high-low agglomeration and low-low agglomeration, but there is no obvious spatial correlation, that is, TE is not affected by neighboring regions. The TE of the eight regions is also different. The average value of TE in the middle reaches of the Yellow River, the eastern coast and the southwest is higher, while the average TE in the south coast, the middle reaches of the Yangtze River and the northeast is lower.(3)From the point of view of input–output fluctuation, the changes of environmental resource inputs, tourism resource inputs and tourism infrastructure construction have the greatest impact on the TE of provinces, and there is a serious redundancy in tourism fixed asset investment. The indicators of the most important and most redundant in each region are basically consistent with the national level.(4)From the perspective of driving factors, economic factors are not the biggest driving force of TE. The development of tertiary industry, urbanization and the improvement of the level of marketization in each province can all drive the promotion of TE in this region.(5)From the perspective of comparative analysis of the results, the TE results obtained on the basis of improving the existing input–output indicators have a significant correlation with other authoritative results, and the results of this paper are scientific and credible.

## Figures and Tables

**Figure 1 ijerph-19-10118-f001:**
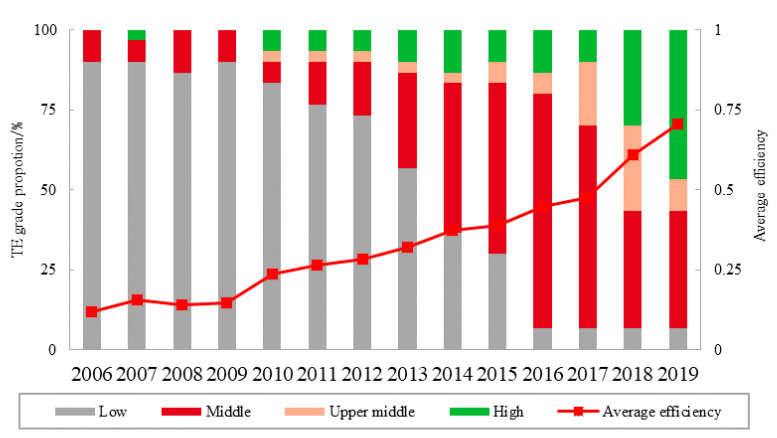
Tourism efficiency mean and grade distribution.

**Figure 2 ijerph-19-10118-f002:**
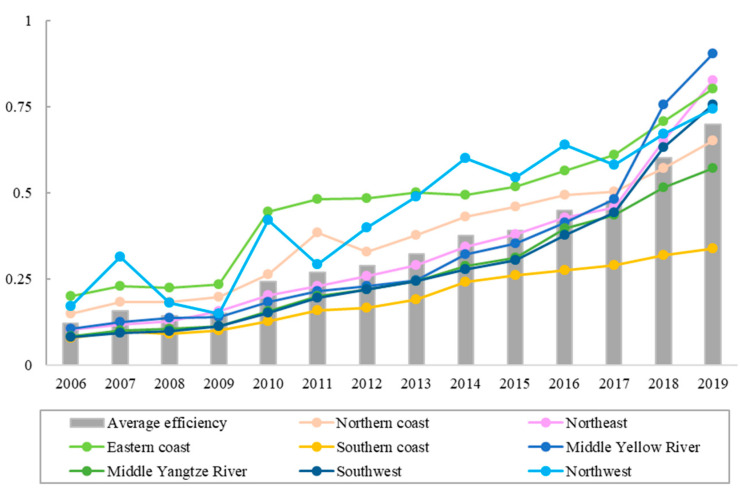
Changes in tourism efficiency in eight regions.

**Figure 3 ijerph-19-10118-f003:**
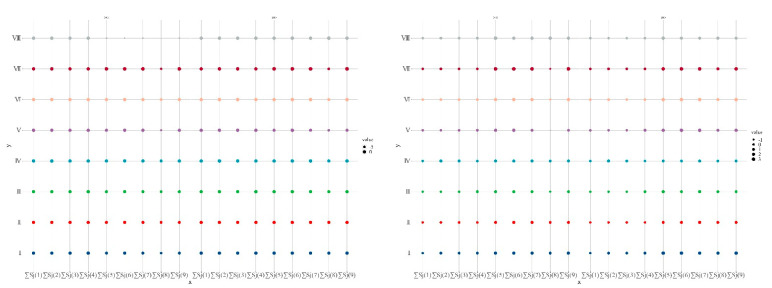
Relative efficiency analysis of composite DEA—cumulative value matrix bubble plot.

**Figure 4 ijerph-19-10118-f004:**
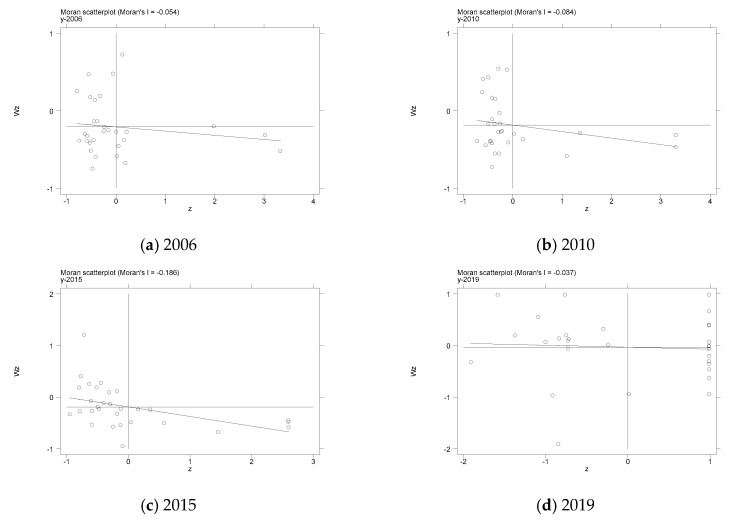
Moran scatter plot of tourism efficiency.

**Table 1 ijerph-19-10118-t001:** Tourism efficiency metrics.

Indicator Category		Specific Indicators		Calculation Method
Input	Economic input	Tourism fixed assets *X*1		The sum of the original value of fixed asset investments in the three pillar industries of tourism (travel agencies, star-rated hotels and tourist scenic spots).
		Tourism infrastructure index *X*2		The quantitative indicators of cultural center institutions, public library institutions, museum institutions, star-rated hotels and travel agencies, public vehicles, civil transport ships, civil transport airports and railway mileage are selected, and the range standardization method and entropy method are used to measure the index.
	Social input	The number of tourism practitioners *X*3		The total number of employees in the three pillar industries of tourism (travel agencies, star-rated hotels and tourist scenic spots).
	Environ-mental input	Environmental resource input index *X*4		The “top-down” method based on energy terminal is used to calculate the energy consumption of key areas (tourism transportation, tourism accommodation, tourism activities) [32,33,34]. The water footprint method is used to calculate the tourism water resources consumption of key links (accommodation, catering, sightseeing activities) [35,36], and the range standardization method and entropy method are used to measure the index.
		Tourism resource endowment index *X*5		The index is measured by range standardization method and entropy method for the number of nature reserves, national scenic spots and famous historical and cultural cities.
Output	Desirable output	Economic output	Total tourist arrivals *Y*1	The sum of total domestic tourist arrivals and total inbound tourist arrivals
			Total tourism revenue *Y*2	The sum of total domestic tourism revenue and total inbound tourism revenue.
		Social output	Urban–rural resident income gap index *Y*3	The ratio of the per capita disposable income of urban residents to the per capita disposable income of rural residents. Before 2015, the per capita disposable income of rural residents was replaced by the per capita net income of farmers. As the expected output, this paper takes the reciprocal treatment of the urban–rural income gap.
	Undesirable output	Environmental output	Tourism pollution emission index *Y*4	The “top-down” method based on energy terminal was used to calculate the tourism carbon emission [32,33,34], and the water footprint method was used to calculate the tourism water resources consumption [35,36]. The tourism solid waste emission was calculated by “per capita domestic waste production × number of tourists × average residence time”, and the index was measured by range standardization method and entropy method.

Note: tourism fixed assets and total tourism income are treated at constant prices to eliminate the influence of price factors in different years.

**Table 2 ijerph-19-10118-t002:** Moran’s I index of tourism efficiency.

Year	I	z	*p*-Value
2006	−0.054	−0.174	0.431
2007	−0.050	−0.185	0.427
2008	−0.103	−0.602	0.273
2009	−0.025	0.084	0.466
2010	−0.084	−0.456	0.324
2011	−0.078	−0.419	0.338
2012	−0.120	−0.774	0.220
2013	−0.150	−1.015	0.155
2014	−0.186	−1.293	0.098
2015	0.186	−1.293	0.098
2016	0.208	−1.459	0.072
2017	−0.224	−1.582	0.057
2018	−0.089	−438	0.331
2019	−0.037	−0.024	0.490

**Table 3 ijerph-19-10118-t003:** Descriptive statistics of variables related to tourism efficiency drivers.

Drivers	Variable Name	Symbol	Calculation Formula	Mean	Std. Dev.	Min	Max
Dependent variable	Tourism efficiency	TE	Undesirable-SBM–DEA	0.36	0.275	0.043	1
Regional economic level	GDP per capita	gdppc	GDP/resident population (RMB/person)	10.339	0.637	7.39	12.502
Industrial structure	The proportion of the tertiary industry	pti	The added value of the tertiary industry/GDP (%)	44.403	9.616	28.303	83.521
Quality of fiscal revenue	Share of tax revenue	str	Tax revenue/general public budget revenue (%)	74.849	8.415	53.000	98.767
Level of opening to the outside world	The proportion of total import and export of goods in GDP	pieg	Total import and export of goods/GDP (%)	27.639	32.524	0.165	166.025
Degree of marketization	Market index	mi	Refers to the level and degree of regional marketization development [45]	6.238	1.715	2.33	10.92
Urban and rural structure	Urbanization rate	ul	Urban area registered population/total population (%)	54.638	13.579	27.46	89.6
Level of digitization	Internet penetration	ip	Number of Internet access households/Number of households with regular primary population	40.725	18.471	3.779	90.686
Location traffic status	Traffic network density	tnd	Total mileage of road network/area of the area (km/km^2^)	0.614	0.288	0.043	1.253
Incidence of public health events	Incidence of infectious diseases	iid	Incidence of infectious diseases/resident population (1/100,000)	260.098	103.000	102.480	738.190
Technological innovation investment	R&D spending intensity	rdi	R&D expenditure/GDP (%)	1.498	1.079	0.197	6.31

**Table 4 ijerph-19-10118-t004:** Regression results of tourism-efficiency drivers.

Explanatory Variables	Pooled-OLS		FE-Robust			RE-Robust
	Coef.	t	Coef.	t	Coef.	z
lngdppc	0.0044 (0.0521)	0.08	0.0035 (0.0340)	0.10	0.0185 (0.0310)	0.60
pti	0.0078 ** (0.0033)	2.36	0.0117 *** (0.0033)	3.58	0.0112 ** (0.0031)	3.62
str	−0.0050 (0.0036)	−1.41	−0.0013 (0.0033)	−0.38	−0.0045 (0.0028)	−1.61
pieg	−0.0010 (0.0012)	−0.79	−0.0018 ** (0.0009)	−0.23	−0.0011 (0.0008)	−1.35
mi	−0.0396 * (0.0186)	−2.13	0.0020 * (0.0163)	0.15	−0.0021 (0.0123)	0.17
ul	0.0149 ** (0.0045)	3.28	0.0204 *** (0.0072)	2.82	0.0147 *** (0.0042)	3.51
ip	0.0024 (0.0020)	1.16	0.0013 (0.0020)	0.64	0.0024 (0.0016)	1.53
lntnd	0.3052 ** (0.1148)	2.66	0.0509 (0.1185)	0.43	0.0790 (0.1060)	0.75
iid	−0.0002 (0.0002)	−0.77	−0.0001 (0.0003)	−0.23	−0.0001 (0.0002)	−0.51
rdi	−0.1659 *** (0.0364)	−4.56	−0.0384 (0.0534)	−0.72	−0.1523 *** (0.0499)	−3.05
-cons	−0.1088	−0.21	−1.1612	−2.63	−0.6623	−1.78
R^2^	0.524		0.662		0.647	
Pooled-OLS vs. FE (F-test)				F = 22.91, Prob > F = 0.0000		
Pooled-OLS vs. RE (LM-test)				LM(Var(u) = 0, lambda = 0) = 668.01, Pr > chi^2^(2) = 0.0000		
FE vs. RE (Hausman test)				F = 58.36, Prob > F = 0.0000		

Note: * *p* < 0.1, ** *p* < 0.05, *** *p* < 0.01.

## Data Availability

The data used to support the findings of this study will be available from the corresponding authors upon request.

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
