# Peer review of "Evolution and Driving Factors of the Spatiotemporal Pattern of Tourism Efficiency at the Provincial Level in China Based on SBM–DEA Model"

_ijerph, 2022, doi:10.3390/ijerph191610118_

Round 1
Reviewer 1 Report
Under the guidance of improving tourism competitiveness and sustainable development, it is particularly important to identify and analyze factors and mechanisms that affect efficiency. Focusing on the current demand for high-quality development of tourism, this paper analyzes the spatiotemporal characteristics and driving factors of tourism efficiency in China with great significance. However, there are still some problems in this paper that need to be discussed with the author. The main problems are as follows:
First of all, The title of the article needs to be changed and perfected. This study is mainly based on the SBM-DEA model to measure the spatiotemporal characteristics and driving factors of China's tourism efficiency. This is one of the important highlights of this article. Therefore, it is recommended to change the title to: Evolution and Driving Factors of the Spatiotemporal Pattern of Tourism Efficiency at the Provincial Level in China Based on SBM—DEA Model.
Secondly, the timeliness of the research of this paper needs to be updated. It is recommended to update the research time period to 2006-2021. On the one hand, this can improve the timeliness of the research of this paper, and on the other hand, this paper can also better reveal the characteristics and driving factors of the spatial and temporal development and changes of tourism efficiency under the impact of the COVID-19 Outbreak.
Third, the construction of the tourism efficiency measurement index system is not comprehensive enough. For example, the comprehensive measurement of the Tourism Infrastructure Index ignores the traffic factor. It is recommended to add indicators such as tourism comprehensive transportation to the comprehensive measurement of the Tourism Infrastructure Index. In addition, tourism has become a necessity for people's happy life, and tourism benefits the people and meets people's new expectations for a better life has become an important value orientation for tourism development. It is worth noting that the "National Tourism and Leisure Development Outline (2022-2030)" also emphasizes in order to improve the quality of tourism products and services, and enrich the connotation of tourism and leisure. Therefore, it is recommended to add the "tourist satisfaction" index to the desirable output index of tourism efficiency measurement, so as to better and objectively measure the tourism efficiency index.
Finally, the selection of factors influencing tourism efficiency is not comprehensive enough. It is recommended to add indicators such as digital economy, finance and taxation, tourism policy, and emergencies, so as to more comprehensively explain the driving factors of China's tourism efficiency. Furthermore,Based on the analysis results, a picture of the driving mechanism of the spatiotemporal variation characteristics of China's tourism efficiency is extracted and constructed.
Author Response
Point 1: The title of the article needs to be changed and perfected. This study is mainly based on the SBM-DEA model to measure the spatiotemporal characteristics and driving factors of China's tourism efficiency. This is one of the important highlights of this article. Therefore, it is recommended to change the title to: Evolution and Driving Factors of the Spatiotemporal Pattern of Tourism Efficiency at the Provincial Level in China Based on SBM—DEA Model.
Response 1: We appreciate for your valuable comment. As you said, the highlight of this article is mainly to measure China's tourism efficiency based on the SBM-DEA model, and then further analyze its spatiotemporal characteristics and driving factors, so the title of this article has been adjusted according to your opinion. Thank you very much for your suggestion.
Title: Evolution and Driving Factors of the Spatiotemporal Pattern of Tourism Efficiency at the Provincial Level in China Based on SBM—DEA Model
Point 2: The timeliness of the research of this paper needs to be updated. It is recommended to update the research time period to 2006-2021. On the one hand, this can improve the timeliness of the research of this paper, and on the other hand, this paper can also better reveal the characteristics and driving factors of the spatial and temporal development and changes of tourism efficiency under the impact of the COVID-19 Outbreak.
Response 2: Thank you for your advice. Since the relevant statistical yearbooks in 2020 and 2021 have not been updated, the lack of indicator data in these two years is serious, and the tourism efficiency value of these two years cannot be calculated. Therefore, this paper controls the research scope in the period of 2006-2019. As you said in your suggestion, the COVID-19 has had a profound impact on tourism, and how to quantify this impact is also a very important and interesting topic. Due to the problem of data acquisition, this paper has not yet quantified it. However, we have added the deficiency of the current research and the prospect of the related research content in the article. It is hoped that when the data are available in the future, we will analyze the impact of COVID-19 on China's tourism efficiency through the relevant data, and explore its influence mechanism. Thank you again for your advice.
4.3. Limitations and future research direction
This paper takes the provincial level as the basic spatial research unit. Future research can expand the research perspective to the level of cities, counties or urban agglomerations, which can provide more valuable references for sustainable tourism development and global tourism. In addition, it is recommended that the national statistics department further refine the statistical caliber of tourism-related data, collect tourism-related statistical data, accelerate the establishment of a tourism satellite account, and support comprehensive research on the sustainable development of China's tourism.
In addition, the emergence of COVID-19 in December 2019 affected more than 200 countries and regions, and had an impact on industries such as health, biomedicine, environment, and tourism. Among them, hotels and tourism activities were the production industries that were severely affected by the epidemic. Therefore, based on detailed data in the future, it is possible to quantitatively analyze the intensity of the impact of COVID-19 on tourism efficiency, explore its impact mechanism on tourism efficiency, and further explore how to improve crisis management capabilities and social governance levels in the face of emergencies.
Point 3: The construction of the tourism efficiency measurement index system is not comprehensive enough. For example, the comprehensive measurement of the Tourism Infrastructure Index ignores the traffic factor. It is recommended to add indicators such as tourism comprehensive transportation to the comprehensive measurement of the Tourism Infrastructure Index. In addition, tourism has become a necessity for people's happy life, and tourism benefits the people and meets people's new expectations for a better life has become an important value orientation for tourism development. It is worth noting that the "National Tourism and Leisure Development Outline (2022-2030)" also emphasizes in order to improve the quality of tourism products and services, and enrich the connotation of tourism and leisure. Therefore, it is recommended to add the "tourist satisfaction" index to the desirable output index of tourism efficiency measurement, so as to better and objectively measure the tourism efficiency index.
Response 3: Thank you for pointing out the problems in the establishment of input indicators in this article. We have added a series of quantitative indicators to the tourism infrastructure index, which are composed of ownership of public vehicles, ownership of civil transport ship, civil transport airport number and railway operating mileage, which can reflect the status of regional tourism transportation infrastructure. At the same time, due to the change of input indicators, the subsequent relevant results and analysis have been revised accordingly, and the revised part has been marked red in the original text. As you said, "tourist satisfaction" is a good indicator to reflect the quality of tourism services, but this indicator needs to be obtained through questionnaires and other forms, and it is difficult to obtain its continuity data under the existing statistical caliber. Therefore, this indicator is not added to the ideal output index of tourism efficiency. Thank you again for your advice.
Table 1. Tourism Efficiency Metrics.
|
Indicator category |
Specific indicators |
Calculation method |
||
|
Input |
Economi-c input |
Tourism fixed assets X1 |
The sum of the original value of fixed assets investment in the three pillar industries of tourism (travel agencies, star level hotels and tourist scenic spots). |
|
|
Tourism Infrastructure Index X2 |
The quantitative indicators of cultural center institutions, public library institutions, museum institutions, star hotels and travel agencies, public vehicles, civil transport ships, civil transport airports and railway mileage are selected, and the range standardization method and entropy method are used to measure the index. |
|||
|
Social input |
The number of tourism practitioners X3 |
The total number of employees in the three pillar industries of tourism (travel agencies, star level hotels and tourist scenic spots). |
||
|
Environ-mental input |
Environmental Resource Input Index X4 |
The "top-down" method based on energy terminal is used to calculate the energy consumption of key areas (tourism transportation, tourism accommodation, tourism activities) [32-34]. The water footprint method is used to calculate the tourism water resources consumption of key links (accommodation, catering, sightseeing activities) [35,36], and the range standardization method and entropy method are used to measure the index. |
||
|
Tourism resource endowment index X5 |
The index is measured by range standardization method and entropy method for the number of nature reserves, national scenic spots and famous historical and cultural cities. |
|||
|
Output |
Desirable output |
Economic output |
Total tourist arrivals Y1 |
The sum of total domestic tourist arrivals and total inbound tourist arrivals |
|
Total tourism revenue Y2 |
The sum of total domestic tourism revenue and total inbound tourism revenue. |
|||
|
Social output |
Urban-rural Resident Income Gap Index Y3 |
The ratio of the per capita disposable income of urban residents to the per capita disposable income of rural residents. Before 2015, the per capita disposable income of rural residents was replaced by the per capita net income of farmers. As the expected output, this paper takes the reciprocal treatment of the urban-rural income gap. |
||
|
Undesira-ble oputput |
Environment-al output |
Tourism Pollution Emission Index Y4 |
The "top-down" method based on energy terminal was used to calculate the tourism carbon emission [32-34], and the water footprint method was used to calculate the tourism water resources consumption [35,36]. The tourism solid waste emission was calculated by "per capita domestic waste production × number of tourists × average residence time", and the index was measured by range standardization method and entropy method. |
|
Point 4: The selection of factors influencing tourism efficiency is not comprehensive enough. It is recommended to add indicators such as digital economy, finance and taxation, tourism policy, and emergencies, so as to more comprehensively explain the driving factors of China's tourism efficiency. Furthermore,Based on the analysis results, a picture of the driving mechanism of the spatiotemporal variation characteristics of China's tourism efficiency is extracted and constructed.
Response 4: Thanks for your valuable comments. Based on your suggestion, we have added "Share of tax revenue" that can reflect the quality of fiscal revenue, "Internet penetration" that can reflect the level of digitization, and "Incidence of infectious diseases" that can reflect the incidence of public health events to the tourism efficiency drivers. At the same time, according to the results of the re-measurement, the analysis of the driving factors and driving mechanism of tourism efficiency has been revised and improved. Thank you again for your advice.
This paper analyzes the driving mechanism of tourism efficiency from 10 aspects such as regional economic level, quality of fiscal revenue, level of opening to the outside world, degree of marketization, urban and rural structure, level of digitization, location traffic status and incidence of public health events and technological innovation investment, and selects key indicators to measure the main driving factors of tourism efficiency.
Table 3. Descriptive Statistics of Variables Related to Tourism Efficiency Drivers.
|
Drivers |
Variable name |
Symbol |
Calculation formula |
Mean |
Std.Dev. |
Min |
Max |
|
Dependent variable |
Tourism efficiency |
TE |
Undesirable-SBM-DEA |
0.36 |
0.275 |
0.043 |
1 |
|
Regional economic level |
GDP per capita |
gdppc |
GDP/resident population (RMB/person) |
10.339 |
0.637 |
7.39 |
12.502 |
|
Industrial structure |
The proportion of the tertiary industry |
pti |
The added value of the tertiary industry/GDP (%) |
44.403 |
9.616 |
28.303 |
83.521 |
|
Quality of fiscal revenue |
Share of tax revenue |
str |
Tax revenue/general public budget revenue (%) |
74.849 |
8.415 |
53.000 |
98.767 |
|
Level of opening to the outside world |
The proportion of total import and export of goods in GDP |
pieg |
Total import and export of goods/GDP (%) |
27.639 |
32.524 |
0.165 |
166.025 |
|
Degree of marketization |
Market index |
mi |
Refers to the level and degree of regional marketization development [45] |
6.238 |
1.715 |
2.33 |
10.92 |
|
Urban and rural structure |
Urbanization rate |
ul |
Urban area registered population/total population (%) |
54.638 |
13.579 |
27.46 |
89.6 |
|
Level of digitization |
Internet penetration |
ip |
Number of Internet access households/Number of households with regular primary population |
40.725 |
18.471 |
3.779 |
90.686 |
|
Location traffic status |
Traffic network density |
tnd |
Total mileage of road network/area of the area(km/km2) |
0.614 |
0.288 |
0.043 |
1.253 |
|
Incidence of public health events |
Incidence of infectious diseases |
iid |
Incidence of infectious diseases/resident population(1/100,000) |
260.098 |
103.000 |
102.480 |
738.190 |
|
Technological innovation investment |
R&D spending intensity |
rdi |
R&D expenditure/GDP (%) |
1.498 |
1.079 |
0.197 |
6.31 |

Reviewer 2 Report
Dear authors, thank you for sharing your research results with us. As understand, the main scientific contribution of this manuscript lies in introducing the input-output of economy, society and environment into the study of tourism efficiency based on a comprehensive consideration of the input of tourism environmental resources and the unexpected output of environmental pollution, as well as the consideration of social benefits. However, it is necessary to support this with a literature review. This is precisely what the manuscript lacks. The literature review should be critical and support the research and outcome through the literature gap. It also allows for more discussion of the findings in the context of the previous literature, which is also needed.
Good luck.
Author Response
Point 1: Dear authors, thank you for sharing your research results with us. As understand, the main scientific contribution of this manuscript lies in introducing the input-output of economy, society and environment into the study of tourism efficiency based on a comprehensive consideration of the input of tourism environmental resources and the unexpected output of environmental pollution, as well as the consideration of social benefits. However, it is necessary to support this with a literature review. This is precisely what the manuscript lacks. The literature review should be critical and support the research and outcome through the literature gap. It also allows for more discussion of the findings in the context of the previous literature, which is also needed.
Response 1: We appreciate for your valuable comment. We have revised and improved the literature review part, firstly, referring to new publications in the field of tourism research, and improving the introduction part of this paper; secondly, we have comprehensively sorted out and summarized the research directions related to tourism efficiency; finally, We discussed the research content more in the context of the previous literature, pointed out the research gaps and research deficiencies in the existing research, and explained the innovative work made in this paper on the basis of the existing research. Thank you very much for your suggestion.
For any economic entity, improving efficiency is an important condition to maintain its sustainable development [4]. Tourism efficiency has gradually become one of the hot spots in tourism research in recent years [5]. The current research is mainly divided into tourism industry factor efficiency, tourism economic efficiency and tourism ecological efficiency. Different research fields of tourism efficiency involve different tourism destination scales such as countries, provinces, counties and cities, and villages, and mainly focus on the differences in spatial and temporal patterns, evolution characteristics, driving mechanisms and factors affecting efficiency. The non-parametric method discusses the efficiency indicators of pure technical efficiency and scale efficiency, and further analyzes the coupling level of tourism efficiency, tourism development scale and economic devel-opment level from a macro perspective, and analyzes the internal state of the tourism sys-tem from a micro perspective [6]. The research on the factor efficiency of tourism industry mainly focuses on the factors of tourism industry, such as travel agency, tourism scenic spot, tourism accommodation industry, tourism transportation industry and so on [7]. Most of them are carried out from the enterprise level. Enterprises are an important force that cannot be ignored in the development of tourism industry, environmental protection and technological innovation [8]. Tourism economic efficiency is a comprehensive performance measurement for the development of one or more tourism destination units. When considering the maximization of input and output, economic indicators are selected, such as fixed asset investment, actual use of foreign investment and tourism income [9-11]. Tourism eco-efficiency is to obtain the maximum economic output with the least resource consumption and environmental cost in tourism activities, taking into account the input of ecological environment resources and the undesired output of environmental pollution caused by tourism [12-14]. With the passage of time, the study of tourism efficiency has gradually developed to more complex topics, such as influencing factors, spatial effects, driving factors and sustainable development of tourism. Guo et al. (2022) [15] combined tourism efficiency with new geo-graphical technology and used geographic detector model to determine the determinants related to the spatial differentiation of tourism ecological efficiency. Some studies have ex-plored the spatial-temporal characteristics of tourism efficiency through exploratory spa-tial data analysis. Li et al. (2020) [16] adopted a spatial autocorrelation model, which shows that the overall spatial correlation of tourism efficiency in Wuling Mountain area of China is weak. Some scholars have carried out research on tourism ecological security, which is an important direction of sustainable development of tourism destinations. Liu et al. (2022) [17] assessed the spatio-temporal evolution and dynamic characteristics of tourism ecological security, as well as influencing factors and driving mechanisms. In addition, Ma et al (2022) [18] proposed a new framework for measuring the temporal and spatial distribution of cross-provincial tourism demand based on search engine index from a geographical perspective, which combines spatial autocorrelation and Geo detector to identify the spatio-temporal distribution pattern of tourism demand at the provincial level in China and detect its driving mechanism.
Although the study of tourism ecological efficiency has considered the undesirable output of environmental pollution categories such as tourism carbon emissions, most of them consider the undesirable output narrowly, ignoring social input and social output [19-21]. It is not in line with the vision of economic, social, environmental sustainable development and high-quality development of tourism destinations. Although previous studies on the determinants of tourism efficiency have made substantial contributions to tourism literature, little attention has been focused on the macro external determinants associated with efficiency, such as economic, social, technical, and institutional factors. At the same time, most of the perspectives of these studies focus on the analysis of specific cases, and they are relatively simple in terms of research scale. In summary, domestic and foreign research results provide certain theoretical support and reference for further re-search on tourism efficiency, but there is still room for improvement in related research. This paper focuses on the current demand for high-quality development of tourism, introduces the input and output of economy, society and environment into the study of tourism efficiency, on the basis of comprehensively considering the undesirable outputs such as tourism environmental resources input and tourism carbon emissions, tourism water pollution emissions and tourism solid waste production, the social benefits brought by tourism are also taken into account.
Based on this, the input-output index system is constructed, and the SBM-DEA model based on undesirable output, compound DEA model and fixed effects model are used to measure China's provincial tourism efficiency, identify the key indicators of input and output, and analyze the spatial differentiation law of tourism efficiency by exploratory spatial data method, further exploration of the drivers of tourism efficiency. This paper analyzes the driving mechanism of tourism efficiency from 10 aspects such as regional economic level, quality of fiscal revenue, level of opening to the outside world, degree of marketization, urban and rural structure, level of digitization, location traffic status and incidence of public health events and technological innovation investment, and selects key indicators to measure the main driving factors of tourism efficiency.

Reviewer 3 Report
Dear authors,
The manuscript entitled Evolution and Driving Factors of the Spatiotemporal Pattern of Tourism Efficiency at the Provincial Level in China represent valuable study which assess in depth the Data Envelopment analysis. Although the presented work with valuable methodology and results deserve to be considered for publishing in the IJERPH, it still has some minor issues needed to be addressed before this step. Authors should cite more recent studies in the topic, for example:
Liu, D., & Yin, Z. (2022). Spatial-temporal pattern evolution and mechanism model of tourism ecological security in China. Ecological Indicators, 139, 108933.
Ma, X., Yang, Z., & Zheng, J. (2022). Analysis of spatial patterns and driving factors of provincial tourism demand in China. Scientific Reports, 12(1), 1-15.
Author Response
Point 1: The manuscript entitled Evolution and Driving Factors of the Spatiotemporal Pattern of Tourism Efficiency at the Provincial Level in China represent valuable study which assess in depth the Data Envelopment analysis. Although the presented work with valuable methodology and results deserve to be considered for publishing in the IJERPH, it still has some minor issues needed to be addressed before this step. Authors should cite more recent studies in the topic, for example:
Liu, D., & Yin, Z. (2022). Spatial-temporal pattern evolution and mechanism model of tourism ecological security in China. Ecological Indicators, 139, 108933.
Ma, X., Yang, Z., & Zheng, J. (2022). Analysis of spatial patterns and driving factors of provincial tourism demand in China. Scientific Reports, 12(1), 1-15.
Response 1: Thank you for your valuable advice. Citing the latest research on this topic can lead to more useful thinking. Based on your suggestion, we have reviewed and cited the latest publications on the topic of tourism efficiency. In addition to the papers you mentioned such as Liu et al. and Ma et al., we also cite the latest publications of scholars such as Guo et al. and Li et al. in this field. Furthermore, according to the latest research progress in the field of tourism efficiency, we further improve the literature review part, we comprehensively sort out and summarize the existing research direction, and point out the research gaps and deficiencies of the existing research, and also explains the innovative work made in this article on the basis of the existing research. Thank you very much for your suggestion again.
With the passage of time, the study of tourism efficiency has gradually developed to more complex topics, such as influencing factors, spatial effects, driving factors and sustainable development of tourism. Guo et al. (2022) [15] combined tourism efficiency with new geo-graphical technology and used geographic detector model to determine the determinants related to the spatial differentiation of tourism ecological efficiency. Some studies have ex-plored the spatial-temporal characteristics of tourism efficiency through exploratory spa-tial data analysis. Li et al. (2020) [16] adopted a spatial autocorrelation model, which shows that the overall spatial correlation of tourism efficiency in Wuling Mountain area of China is weak. Some scholars have carried out research on tourism ecological security, which is an important direction of sustainable development of tourism destinations. Liu et al. (2022) [17] assessed the spatio-temporal evolution and dynamic characteristics of tourism ecological security, as well as influencing factors and driving mechanisms. In addition, Ma et al (2022) [18] proposed a new framework for measuring the temporal and spatial distribution of cross-provincial tourism demand based on search engine index from a geographical perspective, which combines spatial autocorrelation and Geo detector to identify the spatio-temporal distribution pattern of tourism demand at the provincial level in China and detect its driving mechanism.
Although the study of tourism ecological efficiency has considered the undesirable output of environmental pollution categories such as tourism carbon emissions, most of them consider the undesirable output narrowly, ignoring social input and social output [19-21]. It is not in line with the vision of economic, social, environmental sustainable de-velopment and high-quality development of tourism destinations. Although previous studies on the determinants of tourism efficiency have made substantial contributions to tourism literature, little attention has been focused on the macro external determinants associated with efficiency, such as economic, social, technical, and institutional factors. At the same time, most of the perspectives of these studies focus on the analysis of specific cases, and they are relatively simple in terms of research scale. In summary, domestic and foreign research results provide certain theoretical support and reference for further re-search on tourism efficiency, but there is still room for improvement in related research. This paper focuses on the current demand for high-quality development of tourism, in-troduces the input and output of economy, society and environment into the study of tourism efficiency, on the basis of comprehensively considering the undesirable outputs such as tourism environmental resources input and tourism carbon emissions, tourism water pollution emissions and tourism solid waste production, the social benefits brought by tourism are also taken into account.
Based on this, the input-output index system is constructed, and the SBM-DEA model based on undesirable output, compound DEA model and fixed effects model are used to measure China's provincial tourism efficiency, identify the key indicators of input and output, and analyze the spatial differentiation law of tourism efficiency by exploratory spatial data method, further exploration of the drivers of tourism efficiency. This paper analyzes the driving mechanism of tourism efficiency from 10 aspects such as regional economic level, quality of fiscal revenue, level of opening to the outside world, degree of marketization, urban and rural structure, level of digitization, location traffic status and incidence of public health events and technological innovation investment, and selects key indicators to measure the main driving factors of tourism efficiency.
